# A New Lichenized Fungus, *Lendemeriella luteoaurantia*, with a Key to the Species of *Lendemeriella*

Beeyoung-Gun Lee [1,*] and Jae-Seoun Hur [2]

1    Baekdudaegan National Arboretum, Bonghwa 36209, Republic of Korea
2    Korean Lichen Research Institute, Sunchon National University, Suncheon 57922, Republic of Korea; jshur1@sunchon.ac.kr
*    Correspondence: gitanoblue@koagi.or.kr

**Abstract:** *Lendemeriella luteoaurantia* B.G. Lee is described as a new lichen species from South Korea. The new species is identified by smaller, yellow-orange apothecia, larger ascospores with wider septum width, and the absence of Cinereorufa-green pigment and teloschistin, different from the closest species, *L. aureopruinosa* I.V. Frolov, Vondrák, Arup, Konoreva, S. Chesnokov, Yakovczenko and Davydov in morphology and chemistry. Molecular phylogeny employing internal transcribed spacer (nuITS), nuclear large subunit ribosomal RNA (nuLSU), and mitochondrial small subunit (mtSSU) sequences strongly supports the new species as nonidentical in the genus *Lendemeriella*. A preliminary key is provided to assist in the identification of all 10 species of *Lendemeriella*.

**Keywords:** biodiversity; saxicolous; phylogeny; taxonomy; Teloschistaceae





## 1. Introduction

The genus *Lendemeriella* S.Y. Kondr., previously the '*Caloplaca*' *exsecuta* group, was defined recently from the mother genus *Caloplaca* Th. Fr. [1,2]. *Lendemeriella* is characterized by biatorine apothecia containing anthraquinones, i.e., chemosyndrome A such as parietin (major), emodin (minor), and traces of emodinal, emodic acid, parietinic acid, and fallacinal, golden brown episamma, ascospore septum width >3.5 μm, bacilliform pycnoconidia, Cinereorufa-green pigment in thallus or apothecia, and distribution in arctic-alpine to boreal-montane areas [1–3].

*Lendemeriella* is a genus in the subfamily Caloplacoideae and can be confused with similar genera in the subfamily, i.e., *Bryoplaca* Søchting, Frödén and Arup, *Olegblumia* S.Y. Kondr., Lőkös and Hur, *Rufoplaca* Arup, Søchting and Frödén, and *Pyrenodesmia* A. Massal. Vondrák et al. mentioned the similarity of *Lendemeriella*, *Bryoplaca*, and *Parvoplaca* Arup, Søchting and Frödén in the aspect of their habitat preference to humid alpine areas [1]. However, *Lendemeriella* differs from *Bryoplaca* by substrate preference mainly for rock or bark, biatorine apothecia, and the presence of parietin. *Parvoplaca* is easily distinguishable from *Lendemeriella* as the genus inhabits moss, plant debris, or bark with a poorly developed thallus and belongs to the subfamily Xanthorioideae [4]. Kondratyuk et al. compared *Olegblumia* as the most closely related genus to *Lendemeriella* in the introduction of *Lendemeriella* [2]. However, *Lendemeriella* quite differs from *Olegblumia* by the crustose thallus without lobes, the presence of the chemosyndrome A, and the narrow distribution limited to arctic-alpine and boreal-montane areas. Frolov et al. compared *Lendemeriella* with *Rufoplaca* because they have similar ecology [3]. However, *Lendemeriella* differs from *Rufoplaca* by an ascospore septum width of over 3.5 μm and the presence of episamma (the minute granules in epithecium). *Pyrenodesmia*, previously the *Caloplaca xerica* group, can be compared with *Lendemeriella*. The *C. xerica* group is similar to *Rufoplaca*, and *Pyrenodemia* s. lat. previously included *Olegblumia*, i.e., *O. demissa* (Flot. ex Körb.) S.Y. Kondr., Lőkös, Jung Kim, A.S. Kondr., S.O. Oh and Hur [3,4]. However, *Lendemeriella* differs

from *Pyrenodesmia* by biatorine apothecia with an orange pigment other than Sedifolia-grey, bacilliform pycnoconidia, and the presence of the chemosyndrome A [4].

*Lendemeriella* comprised nine species. Eight species were transferred from *Caloplaca*, i.e., *L. borealis* (Vain.) S.Y. Kondr., *L. dakotensis* (Wetmore) S.Y. Kondr., *L. exsecuta* (Nyl.) S.Y. Kondr., *L. lucifuga* (G. Thor) S.Y. Kondr., *L. nivalis* (Körb.) S.Y. Kondr., *L. reptans* (Lendemer and B.P. Hodk.) S.Y. Kondr., *L. sorocarpa* (Vain.) S.Y. Kondr., and *L. tornoensis* (H. Magn.) S.Y. Kondr. [2]. An additional new species, *L. aureopruinosa*, was added by Frolov et al. [3]. Frolov et al. also suggested *Caloplaca lacinulata* (Hue) Zahlbr. and *C. hexaspora* (Hue) T. Okamoto as probable species for *Lendemeriella* [3].

This study describes an additional new species of the genus. Field surveys of the lichen biodiversity in the main mountains of Korea, i.e., Baekdudaegan, were carried out during the summer of 2022, and 24 specimens in *Lendemeriella* were collected on a shaded scree of a valley and the summit of a mountain (Figure 1). The specimens were comprehensively analyzed and did not correspond to any previously known species. We describe them as a new species, *Lendemeriella luteoaurantia*, also providing a preliminary key for all *Lendemeriella* species.

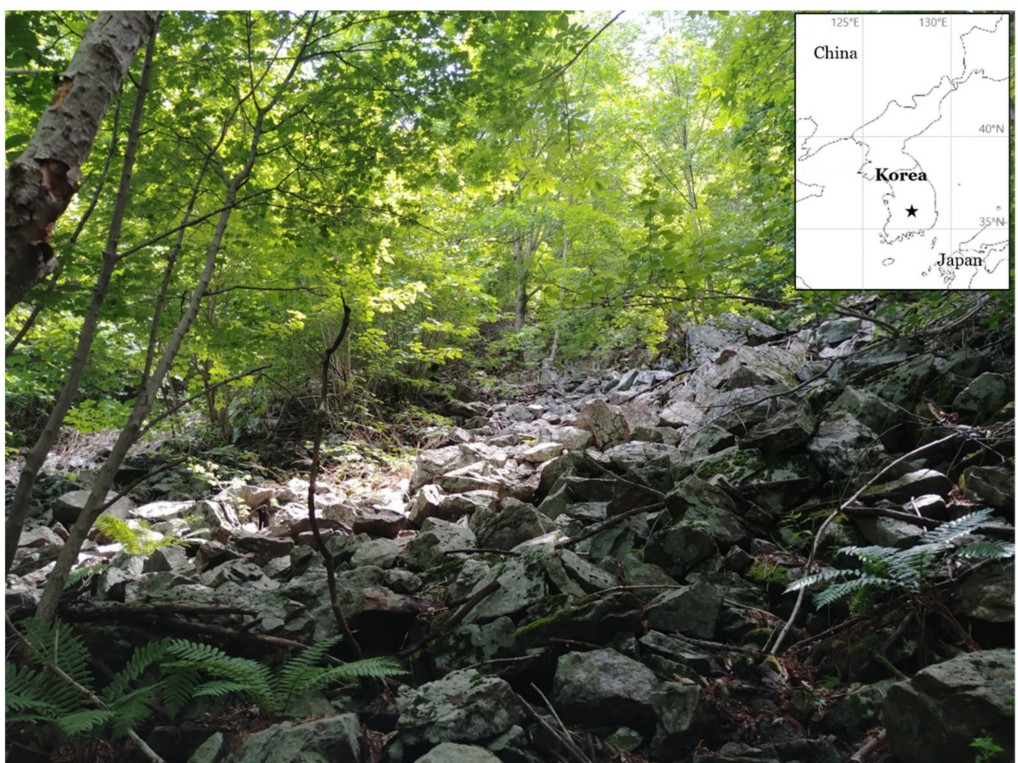

**Figure 1.** Collection site for the new species *Lendemeriella luteoaurantia* (black star mark).

## 2. Materials and Methods

### 2.1. Morphological and Chemical Analyses

Hand sections were prepared manually with a razor blade under a stereomicroscope (Leica LED2500; Leica, Wetzlar, Germany), scrutinized under a compound microscope (Nikon Eclipse E400; Nikon, Tokyo, Japan), and photographed using a software program (NIS-Elements D; Nikon, Tokyo, Japan) and a DS-Fi3 camera (Nikon, Tokyo, Japan) mounted on a Nikon Eclipse Ni-U microscope (Nikon, Tokyo, Japan). The ascospores were examined at 1000× magnification in water. The length and width of the ascospores were measured and the range of spore sizes was shown with average, standard deviation (SD), length-to-width ratio, and number of measured spores. Thin-layer chromatography (TLC) was performed using solvent systems A and C according to standard methods [5]. The

specimens are used in the herbarium of the Baekdudaegan National Arboretum (KBA), South Korea.

## 2.2. Isolation, DNA Extraction, Amplification, and Sequencing

Hand-cut sections of 10 to 20 ascomata per collected specimen were prepared for DNA isolation and DNA was extracted with a NucleoSpin Plant II Kit in line with the manufacturer's instructions (Macherey-Nagel, Düren, Germany). PCR amplification for the internal transcribed spacer region (ITS1-5.8S-ITS2 rDNA) and the nuclear large subunit ribosomal RNA genes was achieved using Bioneer's AccuPower PCR Premix (Bioneer, Daejeon, Republic of Korea) in 20 μL tubes with 16 μL of distilled water, 2 μL of DNA extracts, and 2 μL of primers ITS5 and ITS4 [6] or LR0R and LR5 [7]. The PCR thermal cycling parameters used were 95 °C (15 s), followed by 35 cycles of 95 °C (45 s), 54 °C (45 s), and 72 °C (1 min), and a final extension at 72 °C (7 min) based on Ekman [8]. The annealing temperature was changed by ±1 degree for a better result. PCR purification and DNA sequencing were carried out by the Macrogen genomic research company (Seoul, Republic of Korea).

## 2.3. Phylogenetic Analyses

All nuITS, nuLSU, and mtSSU sequences were aligned and edited manually using ClustalW in Bioedit V7.7.1 [9]. All missing and ambiguously aligned data and phylogenetically uninformative positions were removed and only phylogenetically informative regions were finally analyzed. The final alignment comprised 1265 (nuITS), 2102 (nuLSU), and 965 (mtSSU) columns. In them, variable regions were 265 (nuITS), 532 (nuLSU), and 305 (mtSSU). The phylogenetically informative regions were 734 (nuITS), 638 (nuLSU), and 307 (mtSSU). A concatenation was carried out for combining all nuITS, nuLSU, and mtSSU gene loci. They were manually combined for the informative regions. Four problematic sequences were removed when conflicting results occurred in the internal branches with the bootstrap values $\geq$ 70% and the posterior probabilities $\geq$ 95% in the concatenated tree. Phylogenetic trees with bootstrap values were obtained in RAxML GUI 2.0.6 [10] using the maximum likelihood method with a rapid bootstrap with 1000 bootstrap replications and GTR GAMMA (SYM + I + G4) for the substitution matrix as the best models produced by the model test in the software. The posterior probabilities were obtained in BEAST 2.7.4 [11] using the GTR 123141 model as the appropriate model of nucleotide substitution produced by the Bayesian model averaging methods with bModelTest [12], empirical base frequencies, gamma for the site heterogeneity model, four categories for gamma, and a 10,000,000 Markov chain Monte Carlo chain length with a 10,000-echo state screening and 1000 log parameters. Then, a consensus tree was constructed in TreeAnnotator 2.7.4 [11] with no discard of burnin, no posterior probability limit, a maximum clade credibility tree for the target tree type, and median node heights. All trees were displayed in FigTree 1.4.4 [13] and edited in Microsoft Paint. Overall analyses in the materials and methods were accomplished based on Lee and Hur [14].

## 3. Results and Discussion

### 3.1. Phylogenetic Analyses

The new species is positioned in the *Lendemeriella*-clade in the concatenated tree. The tree describes *L. luteoaurantia*, the new species, being nested with *L. aureopruinosa*, supported by a bootstrap value of 100 and a posterior probability of 1.00 for the branch (Figure 2). Although neighboring *L. aureopruinosa*, *L. luteoaurantia* is located in another branch distant from *L. aureopruinosa*. The new species is included in the subfamily Caloplacoideae and positioned closer to the genera *Bryoplaca*, *Pyrenodesmia*, *Rufoplaca* and *Usnochroma* Søchting, Arup and Frödén than other genera such as *Caloplaca* s. str., *Leproplaca* (Nyl.) Nyl., and *Seirophora* Poelt in the subfamily. This result on the phylogeny for *Lendemeriella* and closely related genera corresponds to Frolov et al. [3]. Their phylogeny also describes *Lendemeriella* neighboring with *Pyrenodesmia*, *Rufoplaca*, and *Usnochroma*, and other genera such as *Caloplaca* s. str., *Leproplaca*, and *Seirophora* are located in a different clade.

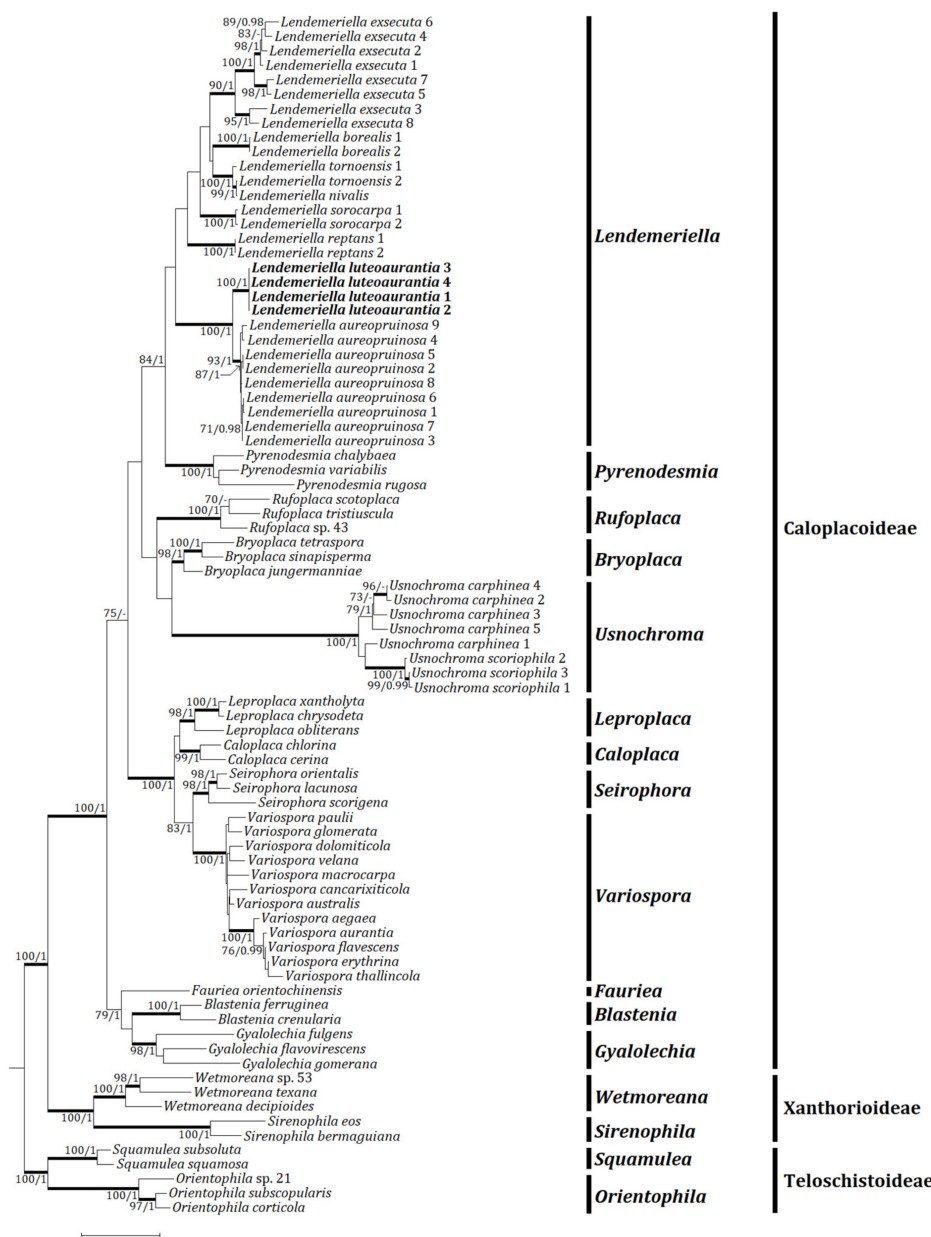

**Figure 2.** Phylogenetic relationships among available species in the genus *Lendemeriella* under the subfamily Caloplacoideae based on a maximum likelihood analysis of the concatenated dataset of nuITS, nuLSU, and mtSSU sequences. The tree was rooted with the sequences of the subfamilies Teloschistoideae and Xanthorioideae based on Arup et al. [4]. Maximum likelihood bootstrap values ≥ 70% and posterior probabilities ≥ 95% are shown above internal branches. Branches with bootstrap values ≥ 90% are shown as fatty lines. The new species, *L. luteoaurantia*, is presented in bold as its DNA sequences were produced from this study. All species names are referred to Table 1.

**Table 1.** Species list and DNA sequence information employed for phylogenetic analysis.

| Species | nuITS | nuLSU | mtSSU | Voucher |
|---|---|---|---|---|
| *Blastenia crenularia* | KC179415 | KC179162 | KC179492 | Søchting 7523 |
| *Blastenia ferruginea* | KC179416 | KC179163 | KC179493 | Søchting 9996 |
| *Bryoplaca jungermanniae* | KC179420 | MT952895 | MT952925 | Søchting 10451 |

**Table 1.** *Cont.*

| Species | nuITS | nuLSU | mtSSU | Voucher |
|---|---|---|---|---|
| *Bryoplaca sinapisperma* | KC179421 | MT952896 | KC179495 | Arup L08184 |
| *Bryoplaca tetraspora* | KC179422 | MT952897 | KC179496 | Søchting 7979 (ITS); Søchting 10,480 (LSU, SSU) |
| *Caloplaca cerina* | KC179425 | KC179168 | KC179499 | Elvebakk 03:084 |
| *Caloplaca chlorina* | KC179426 | KC179169 | KC179500 | Frödén 1876 (ITS); Søchting 7321 (LSU, SSU) |
| *Fauriea orientochinensis* | KX793097 | KX793100 | KX793103 | KoLRI 013957 |
| *Gyalolechia fulgens* | KC179440 | KC179199 | KC179533 | Poelt, Nimis and Tretiach 95/460 (ITS); Arup L06206 (LSU); Søchting 10,586 (SSU) |
| *Gyalolechia gomerana* | KC179441 | KC179200 | KC179534 | Søchting 9653 |
| *Gyalolechia flavovirescens* | AF353966 | KC179198 | KC179532 | Arup L97253 (ITS); Søchting 8648 (LSU, SSU) |
| *Lendemeriella aureopruinosa* 1 | MG954212 | - | - | Chesnokov_sn |
| *Lendemeriella aureopruinosa* 2 | MG954213 | - | - | Chesnokov 239 |
| *Lendemeriella aureopruinosa* 3 | MG954214 | - | - | Chesnokov230 |
| *Lendemeriella aureopruinosa* 4 | MN814228 | MW227504 | MW227332 | LE-L15207 |
| *Lendemeriella aureopruinosa* 5 | MN814229 | MW227511 | MW227333 | LE-L15209 |
| *Lendemeriella aureopruinosa* 6 | MN814230 | - | - | LE-L15210 |
| *Lendemeriella aureopruinosa* 7 | MN814231 | - | - | LE-L15212 |
| *Lendemeriella aureopruinosa* 8 | MN814233 | - | - | Frolov 2232 |
| *Lendemeriella aureopruinosa* 9 | MN814234 | MW227505 | MW227331 | Frolov 2236 (ITS); Frolov 2473 (LSU, SSU) |
| *Lendemeriella borealis* 1 | KX216687 | - | - | IF 1184 |
| *Lendemeriella borealis* 2 | MW227317 | MW227512 | MW227334 | Frolov 2476 |
| *Lendemeriella exsecuta* 1 | MG954130 | - | - | Vondrák 11105 |
| *Lendemeriella exsecuta* 2 | MG954131 | - | - | Vondrák 11110 |
| *Lendemeriella exsecuta* 3 | MG954211 | - | - | Tønsberg 46194 |
| *Lendemeriella exsecuta* 4 | MG954223 | - | - | Zhdanov_sn |
| *Lendemeriella exsecuta* 5 | MG954224 | - | - | Spribille 39677 |
| *Lendemeriella exsecuta* 6 | MG954225 | - | - | Vondrák 7420 |
| *Lendemeriella exsecuta* 7 | MG954226 | - | - | Vondrák 6201 |
| *Lendemeriella exsecuta* 8 | MG954227 | - | - | Spribille 24441 |
| ***Lendemeriella luteoaurantia* 1** | **OQ981385** | **OQ981381** | - | **KBA-L-0004040** |
| ***Lendemeriella luteoaurantia* 2** | **OQ981386** | **OQ981382** | - | **KBA-L-0004041** |
| ***Lendemeriella luteoaurantia* 3** | **OQ981387** | **OQ981383** | - | **KBA-L-0004045** |
| ***Lendemeriella luteoaurantia* 4** | **OQ981388** | **OQ981384** | - | **KBA-L-0004046** |
| *Lendemeriella nivalis* | MG954222 | - | - | Spribille 29306 |
| *Lendemeriella reptans* 1 | JQ686192 | - | JQ686191 | Lendemer 11745 |
| *Lendemeriella reptans* 2 | MH104934 | MH100766 | MH100796 | Lendemer 48186 |
| *Lendemeriella sorocarpa* 1 | MG773658 | - | - | Vondrák 14274 |
| *Lendemeriella sorocarpa* 2 | MG954132 | - | - | Vondrák 12695 |
| *Lendemeriella tornoensis* 1 | MG954220 | - | - | Spribille 26816 |
| *Lendemeriella tornoensis* 2 | MG954221 | - | - | Spribille 29473 |
| *Leproplaca chrysodeta* | KC179448 | KC179206 | MT952933 | Arup L7107 (ITS, LSU); Arup L13261 (SSU) |

**Table 1.** *Cont.*

| Species | nuITS | nuLSU | mtSSU | Voucher |
|---|---|---|---|---|
| *Leproplaca obliterans* | KC179449 | KC179207 | KC179541 | Arup L02331 (ITS, SSU); Arup L03472 (LSU) |
| *Leproplaca xantholyta* | KC179451 | KC179208 | KC179542 | Arup L97278 (ITS); Søchting 9675 (LSU, SSU) |
| *Orientophila corticola* | MN687909 | - | MN687910 | KBA-L-0000118 |
| *Orientophila subscopularis* | KC179375 | - | KC179546 | Frisch Jp171 |
| *Orientophila* sp. 21 | KC179372 | KC179210 | KC179544 | Frisch Jp99 |
| *Pyrenodesmia chalybaea* | KC179454 | MT952921 | KC179571 | Søchting 9351 |
| *Pyrenodesmia rugosa* | MW832828 | MW832804 | MW832825 | KBA-L-0001099 |
| *Pyrenodesmia variabilis* | AF353963 | KC179234 | KC179572 | Arup s.n. (ITS); Arup L03134 (LSU, SSU) |
| *Rufoplaca scotoplaca* | KC179457 | KC179235 | KC179573 | Arup L10032 |
| *Rufoplaca* sp. 43 | KC179458 | KC179236 | KC179574 | Arup L09201 |
| *Rufoplaca tristiuscula* | KC179460 | KC179237 | KC179575 | Arup L08171 |
| *Seirophora lacunosa* | KC179465 | KC179243 | KC179582 | Moberg & Nordin K18:04 |
| *Seirophora orientalis* | KJ021240 | - | - | KoLRI 011917 |
| *Seirophora scorigena* | KC179466 | KC179244 | KC179583 | S. & B Snogerup 17201 |
| *Sirenophila bermaguiana* | KC179299 | KC179245 | KC179584 | Kondratyuk 20487 |
| *Sirenophila eos* | KC179300 | KC179246 | KC179585 | Kärnefelt 20044702 |
| *Squamulea squamosa* | KC179125 | KC179252 | KC179591 | Kärnefelt AM960105 |
| *Squamulea subsoluta* | AF353954 | KC179253 | KC179592 | Arup L97072 |
| *Usnochroma carphinea* 1 | EU639594 | - | - | A. Terron (LEB 4452) |
| *Usnochroma carphinea* 2 | EU639595 | - | - | E. Gaya 201, X. Llimona & M. De Caceres (BCN 13714) |
| *Usnochroma carphinea* 3 | KC179468 | KC179259 | KC179598 | 1998, Roux s.n. |
| *Usnochroma carphinea* 4 | MZ391142 | - | - | VAL_Lich 31793 |
| *Usnochroma carphinea* 5 | - | JQ301548 | JQ301482 | E. Gaya, S. Fernandez-Brime 542 & X. Llimona (BCN) |
| *Usnochroma scoriophila* 1 | EU639596 | - | - | N. Hladun & A. Gomez-Bolea (BCN) |
| *Usnochroma scoriophila* 2 | JQ301664 | JQ301560 | JQ301496 | P. & B. v.d. Boom 38386 |
| *Usnochroma scoriophila* 3 | KC179469 | KC179260 | KC179599 | 1995, Gomez-Bolea |
| *Variospora aegaea* | EU639597 | - | - | E. Gaya 248 & X. Llimona (BCN) |
| *Variospora aurantia* | KC179470 | KC179261 | KC179600 | 1998, Llimona (ITS, SSU); 2006, Lange (LSU) |
| *Variospora australis* | AY233223 | - | - | Gaya 239 |
| *Variospora cancarixiticola* | EU639608 | - | KT291482 | Llimona and Egea s.n. |
| *Variospora dolomiticola* | KC179471 | KC179262 | KC179601 | Thell SP0514 |
| *Variospora erythrina* | KC179472 | - | - | 1998, Lutzoni s.n. |
| *Variospora flavescens* | KC179473 | KC179263 | KC179602 | Søchting 9601 (ITS); Arup L03060 (LSU, SSU) |
| *Variospora glomerata* | KC179474 | KC179264 | KC179603 | Arup L03119 |
| *Variospora macrocarpa* | AF353956 | - | - | Arup L97306 |
| *Variospora paulii* | EU639606 | - | KT291503 | Gaya 183 |
| *Variospora thallincola* | KC179475 | JQ301563 | KC179604 | Søchting 7481 (ITS); Gaya et al. s.n. (LSU); Arup L92148 (SSU) |

**Table 1.** *Cont.*

| Species | nuITS | nuLSU | mtSSU | Voucher |
|---|---|---|---|---|
| *Variospora velana* | KC179476 | KC179265 | KC179605 | Arup L07194 (ITS); Arup L07123 (LSU, SSU) |
| *Wetmoreana decipioides* | KC179333 | KC179269 | KC179608 | Thor 20768 |
| *Wetmoreana texana* | KC179337 | KC179273 | KC179612 | Søchting 9925 |
| *Wetmoreana* sp. 53 | KC179335 | KC179271 | KC179610 | Frödén 1519 |
| **Overall** | **82** | **49** | **50** | |

DNA sequences that were generated for the new species *Lendemeriella luteoaurantia* in this study are presented in bold. All others were obtained from GenBank. The species names are followed by GenBank accession numbers and voucher information. nuITS, internal transcribed spacer; nuLSU, nuclear large subunit ribosomal RNA; mtSSU, mitochondrial small subunit; Voucher, voucher information.

### 3.2. Taxonomy

3.2.1. *Lendemeriella luteoaurantia* B.G. Lee sp. nov. (Figure 3)

MycoBank: MB 848800

Type: South Korea, North Chungcheong Province, Youngdong, Yonghwa-myeon, Mt. Gakho, a shaded scree slope, 36°04′03.6″ N, 127°50′52.6″ E, 900 m alt., on siliceous rock, 16 June 2022, B.G. Lee and J.M. Kim 2022-001116, with *Aspicilia pseudovulcanica* S.Y. Kondr., Lőkös and Hur, *Caloplaca* sp., *Porpidia albocaerulescens* (Wulfen) Hertel and Knoph (holotype: KBA-L-0004045!; GenBank OQ981387 for ITS, and OQ981383 for LSU).

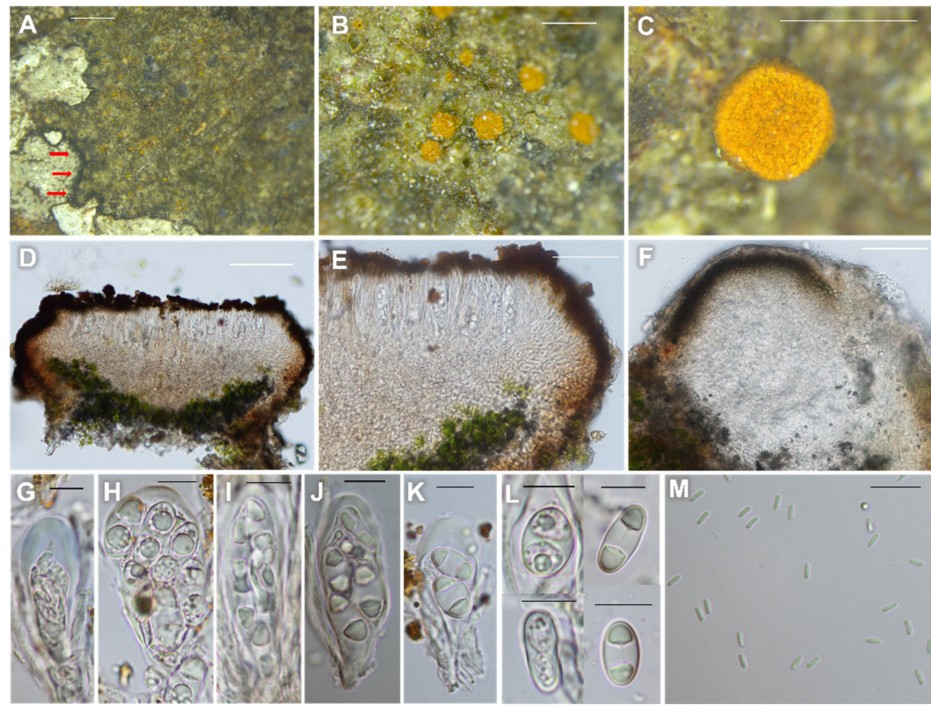

**Figure 3.** *Lendemeriella luteoaurantia* (KBA-L-0004045, holotype) in morphology. (**A–C**) habitus and apothecia. Black prothallus (red arrows) shown in (**A**). (**D,E**) apothecia section showing proper margin only and algal layer present at the base of hypothecium. (**F**) pycnidia. (**G–K**) clavate asci. (**L**) polarilocular ascospores. (**M**) bacilliform pycnoconidia. Scale bars: (**A**) = 2 mm, (**B,C**) = 500 μm, (**D**) = 100 μm, (**E,F**) = 50 μm, (**G–M**) = 10 μm.

Description: thallus saxicolous, crustose, generally continuous but discontinuous or isolated locally, rarely rimose, smooth to slightly rugose, rarely with local thickenings, thin, brown-gray to olive-gray, the margin determinate, 70–150 μm thick; cortex indistinct, up to 10 μm thick; medulla mainly indistinct because of domination of algal layer and substrate-originated large rock crystals dispersed on and below algal layer, up to 150 μm thick;

photobiont coccoid, cells globose to subglobose, 5–15 μm thick, algal layer 60–100 μm thick. Small crystals present between algal cells, dissolving in K. Prothallus black, endosubstratal.

Apothecia numerous, scattered and not concentrated in center, solitary, rarely contiguous, emerging on the surface of thallus, adnate or sessile, slightly constricted at the base, 0.12–0.46 mm diam. (mean = 0.27; SD = 0.08; n = 108). Disc flat or slightly concave when young and flat or slightly convex when mature, with a yellow-orange pruina, yellow-orange to light orange or brownish yellow-orange, c. 200 μm thick; proper margin persistent, prominent, and wide when young and almost even and narrow when mature, entire or crenulate, concolorous with disc, more brownish laterally; thalline margin absent. Parathecium hyaline and brownish to periphery, 35–70 μm wide laterally with epinecral layer c. 10 μm. Epihymenium gold-brown, minutely granular (episamma), K+ wine red and episamma dissolving, 10–15 μm high. Hymenium hyaline, 50–70 μm high. Hypothecium hyaline or slightly yellowish, 70–120 μm high, with algal layer (up to 80 μm) at the base. Oil droplets absent. Paraphyses septate, simple, 1–1.5 μm wide, mainly simple or rarely branched at tips, tips not or slightly swollen, not pigmented, 1–2 μm wide. Asci clavate, 8-spored, 40–55 × 13–25 μm (mean = 50 × 17 μm; SD = 4.8 (L), 3.6 (W); n = 9). Ascospores ellipsoid to widely ellipsoid, 1-septate, polarilocular, permanently hyaline, 11–18 × 5–9.5 μm (mean = 14.7 × 7.6 μm; SD = 1.7 (L), 0.9 (W); L/W ratio 1.4–2.7, ratio mean = 2.0, ratio SD = 0.3; n = 106), septum width 4–6 μm (mean = 5; SD = 0.7; n = 26). Pycnidia dark brown to black, immersed at base and projecting with the upper 1/3, globose and sometimes widening, 135–210 μm high and 160–265 μm wide (SD = 32 (H), 49 (W); n = 5), with brownish wall, K–. Pycnoconidia hyaline, bacilliform, 2.7–4.2 × 0.8–1.4 μm (mean = 3.2 × 1.0 μm; SD = 0.2 (L), 0.1 (W); n = 50).

Chemistry: thallus K–, KC–, C–, Pd–. Apothecia K+ wine red. Epihymenium K+ wine red. Epihymenium and hymenium I+ blue. UV–. Parietin, parietinic acid, fallacinal and emodin were detected by TLC.

Distribution and ecology: The species occurs on siliceous rocks, mainly on the western scree slope shaded by deciduous trees (*Acer pictum* Thunb. var. *mono* (Maxim.) Maxim. ex Franch., *Betula pendula* Roth, *Fraxinus mandshurica* Rupr., *F. rhynchophylla* Hance, and *Magnolia sieboldii* K.Koch), herbs (*Celastrus orbiculatus* Thunb. and *Urtica angustifolia* Fisch. ex Hornem.), and a fern (*Dryopteris crassirhizoma* Nakai). The species is currently known from the shaded scree and the summit of Mt. Gakho (1176 m alt.), South Korea.

Etymology: the species epithet indicates the yellow-orange color of the apothecia.

Notes: the new species is similar to the saxicolous members of the genus, i.e., *L. aureopruinosa*, *L. exsecuta*, and *L. reptans*. However, the new species differs from *L. aureopruinosa* and *L. exsecuta* by pale, small apothecia, colorless hypothecium, little swollen tips of paraphyses, larger ascospores, and the absence of Cinereorufa-green pigment and teloschistin (Table 2).

The new species differs from *L. reptans* by the absence of soredia and thalline margin and the presence of emodin, fallacinal, parietin, and parietinic acid (Table 2).

Without regard to the substrate preference for rock, the new species is similar to the corticolous *L. borealis* in having yellow-orange and small (<0.5 mm diam.) apothecia, black prothallus, epihymenium with K+ red reaction, and the presence of emodin, fallacinal, parietin, and parietinic acid. However, the new species differs from *L. borealis* by darker and thicker thallus, higher hypothecium, larger ascospores with wider septum width, and the absence of teloschstin [15–17].

*Caloplaca lacinulata* is another comparable species that is a probable member of *Lendemeriella* and previously reported from South Korea [3,18]. However, the new species differs from *C. lacinulata* by continuous thallus, crystal-dominated medulla, presence of prothallus, smaller and paler apothecia, higher hypothecium, larger ascospores, larger pycnoconidia, and the absence of lacinules (vegetative propagules) [18,19].

**Table 2.** Comparison of the new species with closely related species in the genus *Lendemeriella*.

| Species | *Lendemeriella luteoaurantia* | *Lendemeriella aureopruinosa* | *Lendemeriella exsecuta* | *Lendemeriella reptans* |
|---|---|---|---|---|
| Soredia | not observed | not observed | not observed | present |
| Apothecia color | yellow-orange to light orange | dark orange to orange-red | brown-yellow, orange-brown, or black | red-brown |
| Apothecia (mm in diam.) | 0.1–0.4 | 0.3–0.6 | 0.2–0.7 | immature |
| Thalline margin | absent | absent | absent | present (gray) |
| Hypothecium color | colorless to slightly yellowish | yellowish (upper), colorless (lower) | yellow to brown | – |
| Paraphysial tip width (μm) | 1–2 | 3–4 | 1.5–4 | – |
| Pigment in true exciple | not pigmented | Cinereorufa-green | Cinereorufa-green | – |
| Ascospores (μm) | 11–18 × 5–9.5 | 11.5–15 × 6–7 | 12–16.5 × 6–7.5 | – |

**Table 2.** *Cont.*

| Species | *Lendemeriella luteoaurantia* | *Lendemeriella aureopruinosa* | *Lendemeriella exsecuta* | *Lendemeriella reptans* |
|---|---|---|---|---|
| Substance | emodin, parietin, parietinic acid, fallacinal | emodin, fallacinal parietin, parietinic acid, teloschistin | 7-chloroemodin, emodin, fallacinal, fragilin, parietin, parietinic acid, teloschistin | no substance |
| Reference | KBA-L-0004040 (isotype), KBA-L-0004041 (isotype), KBA-L-0004045 (holotype), KBA-L-0004046 (isotype) | [3] | [20] | [21] |

The morphological and ecological characteristics for the closely related species are referenced from the previous literature. All information on the new species is produced from selected types of specimens in this study.

Additional specimens examined: South Korea, North Chungcheong Province, Youngdong, Yonghwa-myeon, Mt. Gakho, a shaded scree slope, on siliceous rock, 16 June 2022, B.G. Lee and J.M. Kim 2022-001111, with *Porina leptalea* (Durieu and Mont.) A.L. Sm. (isotype: KBA-L-0004040; GenBank OQ981385 for ITS, and OQ981381 for LSU); same locality, on siliceous rock, 16 June 2022, B.G. Lee and J.M. Kim 2022-00112, with *Aspicilia asteria* Hue (isotype: KBA-L-0004041; GenBank OQ981386 for ITS and OQ981382 for LSU); same locality, on siliceous rock, 16 June 2022, B.G. Lee and J.M. Kim 2022-001114, with *Caloplaca* sp., *Circinaria caesiocinerea* (Nyl. ex Malbr.) A. Nordin, Savić and Tibell, *Porina leptalea* (isotype: KBA-L-0004043); same locality, on siliceous rock, 16 June 2022, B.G. Lee and J.M. Kim 2022-001115, with *Aspicilia pseudovulcanica*, *Porpidia albocaerulescens* (isotype: KBA-L-0004044); same locality, on siliceous rock, 16 June 2022, B.G. Lee and J.M. Kim 2022-001117, with *Circinaria caesiocinerea*, *Porpidia albocaerulescens* (isotype: KBA-L-0004046; GenBank OQ981388 for ITS and OQ981384 for LSU); same locality, on siliceous rock, 16 June 2022, B.G. Lee and J.M. Kim 2022-001119, with *Aspicilia pseudovulcanica*, *Caloplaca* sp., *Lepraria* sp. (isotype: KBA-L-0004048); same locality, on siliceous rock, 16 June 2022, B.G. Lee and J.M. Kim 2022-001122, with *Aspicilia pseudovulcanica*, *Circinaria caesiocinerea*, *Porina leptalea* (isotype: KBA-L-0004051); same locality, on siliceous rock, 16 June 2022, B.G. Lee and J.M. Kim 2022-001123 with *Aspicilia* sp., *Porina leptalea* (isotype: KBA-L-0004052); same locality, on siliceous rock, 16 June 2022, B.G. Lee and J.M. Kim 2022-001125, with *Aspicilia pseudovulcanica*, *Porina leptalea*, *Rimularia geumodoensis* (S.Y. Kondr., Lőkös and Hur) S.Y. Kondr., Lőkös, and Hur (isotype: KBA-L-0004054); same locality, on siliceous rock, 16 June 2022, B.G. Lee and J.M. Kim 2022-001129, with *Caloplaca* sp., *Porpidia albocaerulescens*, *Pseudosagedia guentheri* (Flot.) Hafellner and Kalb (isotype: KBA-L-0004058); same locality, on siliceous rock, 16 June 2022, B.G. Lee and J.M. Kim 2022-001137, with *Aspicilia pseudovulcanica*, *Circinaria caesiocinerea*, *Porina leptalea* (isotype: KBA-L-0004066); same locality, on siliceous rock, 16 June 2022, B.G. Lee and J.M. Kim 2022-001142, with *Aspicilia pseudovulcanica*, *Porina*

*leptalea* (isotype: KBA-L-0004071); same locality, on siliceous rock, 16 June 2022, B.G. Lee and J.M. Kim 2022-001143, with *Gyalolechia flavovirescens* (Wulfen) Søchting, Fröden and Arup, *Porina leptalea*, *Pseudosagedia guentheri*, *Rimularia geumodoensis*, *Verrucaria* sp. (isotype: KBA-L-0004072); same locality, on siliceous rock, 16 June 2022, B.G. Lee and J.M. Kim 2022-001144, with *Aspicilia pseudovulcanica*, *Caloplaca* sp., *Circinaria caesiocinerea*, *Porina leptalea*, *Rimularia geumodoensis* (isotype: KBA-L-0004073); same locality, on siliceous rock, 16 June 2022, B.G. Lee and J.M. Kim 2022-001150, with *Circinaria caesiocinerea*, *Ionaspis lacustris* (With.) Lutzoni (isotype: KBA-L-0004079); same locality, on siliceous rock, 16 June 2022, B.G. Lee and J.M. Kim 2022-001151, with *Aspicilia asteria*, *A. pseudovulcanica*, *Ionaspis lacustris*, *Porpidia albocaerulescens*, *Rimularia geumodoensis* (isotype: KBA-L-0004080); same locality, on siliceous rock, 16 June 2022, B.G. Lee and J.M. Kim 2022-001152, with *Caloplaca* sp. (isotype: KBA-L-0004081); same locality, on siliceous rock, 16 June 2022, B.G. Lee and J.M. Kim 2022-001153, with *Aspicilia pseudovulcanica*, *Caloplaca* sp., *Circinaria caesiocinerea*, *Pseudosagedia guentheri* (isotype: KBA-L-0004082); same locality, on siliceous rock, 16 June 2022, B.G. Lee and J.M. Kim 2022-001154, with *Porina leptalea*, *Rinodina oxydata* (A. Massal.) A. Massal. (isotype: KBA-L-0004083); same locality, on siliceous rock, 16 June 2022, B.G. Lee and J.M. Kim 2022-001155, with *Circinaria caesiocinerea*, *Gyalolechia flavovirescens*, *Porina leptalea* (isotype: KBA-L-0004084); same locality, on siliceous rock, 16 June 2022, B.G. Lee and J.M. Kim 2022-001157, with *Aspicilia pseudovulcanica*, *Circinaria caesiocinerea* (isotype: KBA-L-0004086); same locality, on siliceous rock, 16 June 2022, B.G. Lee and J.M. Kim 2022-001158, with *Circinaria caesiocinerea*, *Gyalolechia flavovirescens*, *Rinodina oxydata* (isotype: KBA-L-0004087); South Korea, North Chungcheong Province, Youngdong, Yonghwa-myeon, Mt. Gakho (summit), 36°03′45.1″ N, 127°50′46.4″ E, 1176 m alt., on siliceous rock, 24 May 2022, B.G. Lee and H.W. Kim 2022-000173 with *Acarospora badiofusca* (Nyl.) Th. Fr., *Aspicilia subgoettweigensis* S.Y. Kondr., Lőkös and Hur, *Buellia* sp., *Lecanora saxigena* Lendemer and R.C. Harris, *Micarea* aff. *peliocarpa* (Anzi) Coppins and R. Sant., *Scoliciosporum umbrinum* (Ach.) Lojka (paratype: KBA-L-0003102).

### 3.2.2. Key to the Species of *Lendemeriella* (Table 3)

The key includes all 10 species in *Lendemeriella*. The genus *Lendemeriella* was defined by Kondratyuk et al. [2] with eight new combinations. Among them, *L. dakotensis* is doubtful for the genus, and the species was not supported with any proof for the new combination. The species differs from other *Lendemeriella* species by subsquamulose and lobed thallus, absence of proper margin in the apothecia, and no secondary metabolites, as well as no reaction to spot test. *Lendemeriella dakotensis* may not be in *Lendemeriella* at last but is included in the key at present.

**Table 3.** Key to the species of *Lendemeriella*.

| | | |
|---|---|---|
| 1. | On rock or moss on rock | **2** |
| – | On bark | **7** |
| 2. | Directly on rock | **3** |
| – | On moss | **6** |
| 3. | Soredia present; apothecia immature; thalline margin gray; no substance | ***L. reptans*** |
| – | Soredia absent; apothecia developed; proper margin present only; emodin, fallacinal, parietin, parietinic acid present or other substance present | **4** |
| 4. | Apothecia yellow-orange to light orange, less than 0.5 mm diam.; hypothecium entirely colorless; tips of paraphyses little swollen; ascospores 11–18 × 5–9.5 μm; Cinereorufa-green pigment absent; teloschistin absent | ***L. luteoaurantia*** |
| 5. | Apothecia brownish or entirely black; 7-chloroemodin and fragilin present; arctic-alpine | ***L. exsecuta*** |
| – | Apothecia reddish; 7-chloroemodin and fragilin absent; boreal-montane | ***L. aureopruinosa*** |
| 6. | On *Racomitrium*; thalline margin pale to dark gray; ascospores 25–30 × 3–5 μm; distributed in mid to high latitudes | ***L. nivalis*** |

**Table 3.** *Cont.*

| | | |
|---|---|---|
| – | Apothecia brown-yellow, red-brown to black, 0.5–0.6 mm diam.; hypothecium yellowish to brownish; tips of paraphyses swollen; ascospores 12–17 × 6–7.5 μm; Cinereorufa-green pigment present in true exciple; teloschistin present | **5** |
| – | On *Andreaea* or *Grimmia*; thalline margin absent; ascospores 17–20 × 6–7.5 μm; limited to high latitudes | *L. tornoensis* |
| 7. | Soredia present; apothecia extremely rare or absent | **8** |
| – | Soredia absent; apothecia generally present | **9** |
| 8. | Soralia K+ purple | *L. lucifuga* |
| – | Soralia K–, C+ pale yellow | *L. sorocarpa* |
| 9. | On deciduous trees (*Alnus*, *Salix* or *Sorbus*); thallus whitish, smooth to wrinkled; prothallus black; apothecia yellow, orange to reddish, 0.2–0.5 mm diam.; thalline margin absent; epihymenium K+ red; emodin, fallacinal, parietin, parietinic acid, and teloschistin present | *L. borealis* |
| – | On conifers; thallus gray-brown, subsquamulose, margin uplifted and lobed; prothallus absent; apothecia brown 0.5–0.8 mm diam.; thalline margin present; epihymenium K–; no substance | *L. dakotensis* |

**Author Contributions:** Conceptualization, B.-G.L.; methodology, B.-G.L. and J.-S.H.; validation, B.-G.L. and J.-S.H.; formal analysis, B.-G.L.; investigation, B.-G.L.; data curation, B.-G.L. and J.-S.H.; writing—original draft preparation, B.-G.L.; writing—review and editing, B.-G.L.; supervision, B.-G.L. and J.-S.H.; project administration, B.-G.L.; funding acquisition, B.-G.L. All authors have read and agreed to the published version of the manuscript.

**Funding:** This work was supported by a grant from the Baekdudaegan National Arboretum (2022-KS-OB-02-01-01) for the biodiversity conservation of the Baekdudaegan mountains of Korea.

**Institutional Review Board Statement:** Not applicable.

**Informed Consent Statement:** Not applicable.

**Data Availability Statement:** Publicly available datasets were analyzed in this study. All newly generated sequences were deposited in GenBank (https://www.ncbi.nlm.nih.gov/genbank/, accessed on 18 May 2023; Table 1). All new taxa were deposited in MycoBank (https://www.mycobank.org/, accessed on 18 May 2023).

**Conflicts of Interest:** The authors declare no conflict of interest.

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
