# Peer review of "A New Lichenized Fungus, Lendemeriella luteoaurantia, with a Key to the Species of Lendemeriella"

_diversity, doi:10.3390/d15070845_

Round 1

Reviewer 1 Report

The paper needs improvement. I have made several comments in the file.

Additional comments:

Authors need to specify why they selected these species (not belonging to Lendemeriella) for the phylogeny.

Notes how to distinguish the genus from other morphologically similar genera must be added and these genera must be characterised. Otherwise, the construction of the key make no sense as there are several species which may potentially resemble Lendemeriella species, including the new taxon. Or if the genus is more like genetically defined, the key should include several other species resembling Lendemeriella.

The description of the tree must be improved and enlarged, including discussion with previous papers.

Under the introduction of new species: notes on the similar species should be enlarged by addition also corticolous members of the genus as well other morphologically similar species of Teloshistaceae.

English needs to be checked. Some parts seem to well-written, but several need improvement!

Reviewer 2 Report

This paper describes a new species of Lendemeriella from S Korea. Although I am not particularly fond of descriptions of species which are known from a single locality, the generic attribution and the distinction toward similar congeneric species are supported by molecular data. The paper is concise, clear and scientifically sound. The English is quite good, but could be improved here and there. I have marked several suggestions on the appached pdf manuscript. On the whole, I think that the paper is worty to be published with a minor revision.

This paper describes a new species of Lendemeriella from S Korea. Although I am not particularly fond of descriptions of species which are known from a single locality, the generic attribution and the distinction toward similar congeneric species are supported by molecular data. The paper is concise, clear and scientifically sound. The English is quite good, but could be improved here and there. I have marked several suggestions on the appached pdf manuscript. On the whole, I think that the paper is worty to be published with a minor revision.

Reviewer 3 Report

This paper provides detailed description of a new species from Korea, with nice pictures and phylogenetic support, but there are several points still need to be improved:

1. In the phylogenetic tree, L. aureopruinosa 1 has large genetic distance fro, other samples, it might be another species, or the sequence quality might be problematic, please recheck this sample.

2. The circumscription of the genus Lendemeriella is problematic, it is not a monophyletic genus, the genus Pyrenodesmia is also clustered in the same clade, it is more reasonable to combine these two genera as one, you should discuss why you accept these two separate genera.

3. Type information is too redundant, the most important information of the type specimen is hard to find in such a long paragraph. And the information of trees is already in the section "Distribution and ecology", which is already enough. You don't need to list information of every lichen samples and GenBank numbers from this same locality, which is not related to this study.

Quality of English language is fine.
